# The Assessment of Sentinel Lymph Node Mapping Methods in Endometrial Cancer

**DOI:** 10.3390/jcm14030676

**Published:** 2025-01-21

**Authors:** Wiktor Szatkowski, Karolina Pniewska, Maja Janeczek, Janusz Ryś, Tomasz Banaś, Konrad Muzykiewicz, Ewa Iwańska, Jerzy Jakubowicz, Kazimierz Karolewski, Agnieszka Szadurska, Paweł Blecharz

**Affiliations:** 1Department of Gynecological Oncology, Maria Skłodowska-Curie National Research Institute, Kraków Branch, 31-115 Kraków, Poland; karolina.pniewska@krakow.nio.gov.pl (K.P.); maja.janeczek@krakow.nio.gov.pl (M.J.); tomasz.banas@krakow.nio.gov.pl (T.B.); konrad.muzykiewicz@krakow.nio.gov.pl (K.M.); ewa.iwanska@krakow.nio.gov.pl (E.I.); jerzy.jakubowicz@krakow.nio.gov.pl (J.J.); kazimierz.karolewski@krakow.nio.gov.pl (K.K.); pawel.blecharz@krakow.nio.gov.pl (P.B.); 2Departament of Pathology, Maria Skłodowska-Curie National Research Institute, Kraków Branch, 31-115 Kraków, Poland; janusz.rys@krakow.nio.gov.pl; 3Departament of Radiotherapy, Maria Skłodowska-Curie National Research Institute, Kraków Branch, 31-115 Kraków, Poland; agnieszka.szadurska@krakow.nio.gov.pl

**Keywords:** sentinel lymph node biopsy, endometrial neoplasms, technetium Tc99m sulfur colloid, indocyanine green, Patent Blue

## Abstract

**Background/Objectives**: Sentinel lymph node biopsy (SLNB) is a minimally invasive technique used to assess lymphatic involvement in endometrial cancer (EC), offering reduced surgical morbidity compared to routine lymphadenectomy. Despite its widespread use, the optimal combination of tracers for SLN detection remains a subject of debate. **Methods**: This retrospective cohort study included 119 patients with early-stage EC treated at the Maria Skłodowska-Curie National Research Institute of Oncology between 2016 and 2021. SLNB was performed using technetium-99m (Tc99m), indocyanine green (ICG), Patent Blue, or combinations of these tracers. Detection rates for unilateral and bilateral SLNs and the accuracy of metastasis identification were analyzed. **Results**: The overall SLN detection rate was 97.5%. Individual tracer detection rates were 100% for ICG, 100% for Patent Blue, and 96% for Tc99m. Combined tracers achieved detection rates of 96.9% (Tc99m and ICG) and 97.3% (Tc99m and Patent Blue). Bilateral detection was highest with Tc99m and ICG (90.6%) and Patent Blue alone (91%). Metastases were identified in 12% of cases, with combined methods improving metastatic detection. Tc99m yielded no “empty nodes”, compared to 1.7% with Patent Blue and 0.8% with ICG. **Conclusions**: While combining Tc99m with dyes did not significantly improve overall SLN detection rates, it enhanced metastatic identification and reduced false-negative results. These findings suggest that combined tracer methods optimize SLNB accuracy in endometrial cancer. Prospective studies are needed to confirm these results.

## 1. Introduction

Endometrial cancer (EC) is among the three most prevalent malignant tumors of the reproductive system in women in developed countries [1]. Over recent years, its incidence has risen, making EC the most common gynecological cancer. This increase is largely attributed to risk factors such as obesity, diabetes, prolonged exposure to unopposed estrogen, and an aging population.

The standard treatment for early-stage EC typically involves a total hysterectomy with bilateral salpingo-oophorectomy and pelvic or para-aortic lymphadenectomy [2]. Lymph node status is a critical determinant of prognosis and guides decisions regarding adjuvant therapy. However, routine lymphadenectomy remains controversial due to the associated risks of complications, including vascular and nerve injuries, lymphedema, and lymphocyst formation [3]. These risks are particularly pronounced in obese patients and those with significant comorbidities, for whom extended surgical procedures can lead to increased perioperative morbidity and prolonged recovery.

Sentinel lymph node biopsy (SLNB) has emerged as a minimally invasive alternative to traditional lymphadenectomy. This procedure enables the detection of lymph node metastases while reducing surgical morbidity [4,5]. SLNB focuses on identifying and removing the sentinel lymph node—the first node to which lymph from the primary tumor drains—thereby allowing for precise cancer staging [6]. The success of SLNB relies on the accurate identification of SLNs, achieving bilateral detection, and minimizing false-negative results.

Comparative studies have shown SLNB to be an effective and safe method for evaluating lymph node status in EC, yielding outcomes comparable to those of complete lymphadenectomy [7,8]. However, for SLNB to replace traditional lymphadenectomy as the standard of care, it must achieve high bilateral detection rates and ensure that the identified nodes contain true lymphatic tissue rather than adipose tissue.

Currently, three primary techniques are employed to identify sentinel lymph nodes (SLNs): radioisotopic tracers (e.g., technetium-99m [Tc99m]), dyes (e.g., Patent Blue), and fluorescent optical tracers (e.g., indocyanine green [ICG]). Each method has distinct strengths and limitations. Tc99m offers excellent tissue penetration and allows preoperative localization through SPECT/CT imaging. Patent Blue is a straightforward, cost-effective technique that does not require specialized equipment. In contrast, ICG provides superior optical resolution using near-infrared fluorescence, making it particularly advantageous in obese patients or those with altered anatomy.

However, each method also has inherent limitations. Tc99m requires access to nuclear medicine facilities, which may not always be available. Patent Blue carries a risk of allergic reactions, including anaphylaxis, while ICG has a short retention time within lymphatic channels, necessitating the precise timing of the procedure. To address these challenges, combining tracers—such as Tc99m with ICG or Tc99m with Patent Blue—has been proposed to enhance detection rates and compensate for individual limitations. Despite this progress, determining the optimal combination of techniques remains an area of active investigation.

The quest for an ideal SLN detection method in EC continues, driven by technical challenges such as “empty nodes”, tracer dye diffusion, and lymphatic channel obstruction caused by tumor emboli [9,10,11,12]. These challenges highlight the need for ongoing procedural refinements. Combining techniques may improve both SLN detection rates and the accuracy of metastasis identification, providing a more robust approach to staging.

The advent of pathological ultrastaging has further advanced SLN evaluation by enhancing the detection of micrometastases and isolated tumor cells (ITCs). This detailed histopathological approach—using serial sectioning and immunohistochemistry—has significant prognostic implications, enabling the identification of low-volume metastases that might otherwise go undetected [13].

The aim of this study was to compare the effectiveness of different SLN detection methods (Tc99m, ICG, and Patent Blue) and evaluate the impact of combining these tracers on overall detection rates, sample quality, and metastasis identification in patients with early-stage EC.

## 2. Materials and Methods

### 2.1. Study Design and Setting

This retrospective cohort study was conducted at the Maria Skłodowska-Curie National Research Institute of Oncology (NIO-PIB), Kraków Branch, and included patients diagnosed with EC who underwent surgical staging, including SLNB, between 2016 and 2021.

Ethical approval for this study was obtained from the Institutional Review Board of the Maria Skłodowska-Curie National Research Institute of Oncology (Approval No. 10/2025, issued on 9 January 2025). Details regarding patient consent and data anonymization are provided in the “Institutional Review Board Statement” and “Informed Consent Statement” sections of this manuscript.

As this is a retrospective cohort study, a formal power analysis to calculate the required sample size was not performed. The sample size was determined based on the number of eligible patients with complete medical records who met the inclusion criteria during the study period.

### 2.2. Inclusion and Exclusion Criteria

Patients were included in this study if they had clinically early-stage endometrial cancer (FIGO stage I–II, 2009 classification) [14,15], had not received prior neoadjuvant therapy such as chemotherapy or radiotherapy, and had complete clinical and histopathological data available for analysis.

Exclusion criteria encompassed advanced-stage disease (FIGO stage III–IV), documented allergic reactions to any of the tracers (ICG or Patent Blue), age below 18 or above 85 years, and comorbidities that contraindicated surgical intervention.

### 2.3. Sentinel Lymph Node Identification Procedure

Surgical staging for endometrial cancer included a total hysterectomy with bilateral salpingo-oophorectomy and SLN identification. The SLN mapping was performed using the following methods.

#### 2.3.1. Radioactive Tracer Administration (Tc99m)

Technetium-99m-labeled human albumin colloid (NanoColl, GE Healthcare, Chicago, IL, USA) was utilized as the radioactive tracer. A total dose of 120 MBq Tc99m was injected into the cervix at the 3 and 9 o’clock positions. The injection was divided equally between superficial (2–3 mm depth) and deep (10–15 mm depth) layers of the cervical stroma, using thin-walled 21G needles.

Preoperative imaging was conducted at 5 min, 60 min, and 18 h post-injection to ensure accurate localization of SLNs. Both static scintigraphy and SPECT-CT imaging were performed using AnyScan Mediso equipment, providing detailed preoperative mapping of SLNs to guide intraoperative procedures.

#### 2.3.2. Dye Administration

Two dyes were used for SLN visualization:Indocyanine green (ICG): a solution of 0.5 mL (1.250 mg ICG) diluted in 5 mL of water was injected at the same cervical positions as the radioactive tracer.Patent Blue: a volume of 2 mL (1 mL per injection site) was injected into the cervix.

Dyes were administered 15–30 min before the initiation of surgery to ensure effective localization during the procedure.

#### 2.3.3. Sentinel Lymph Node Identification

SLN identification was performed laparoscopically using the following tools and techniques:Gamma probe (Gamma Finder 2, Word of Medicine): used intraoperatively to detect Tc99m activity within sentinel lymph nodes.VS3 iridium laparoscopic system (Visionsense 3DHD and IR Fluorescence V): enabled the visualization of ICG fluorescence and identification of Patent Blue-stained nodes.

SLNs were anatomically classified into the following regions: obturator, external iliac, internal iliac, common iliac, and para-aortic. Following excision, each SLN was verified ex vivo using the gamma probe to confirm Tc99m activity.

### 2.4. Histopathological Examination

Excised lymph nodes were fixed in 10% formalin and underwent comprehensive histopathological evaluation. Each node was sectioned into 2 mm slices and stained with hematoxylin and eosin (H&E). Immunohistochemistry for cytokeratins was performed to identify metastatic involvement. SLNs were classified based on the size of metastatic lesions as follows:Macrometastases: lesions > 2 mm.Micrometastases: lesions measuring between 0.2 mm and 2 mm.Isolated tumor cells (ITCs): lesions ≤ 0.2 mm.

### 2.5. Statistical Analysis

Data analysis was performed using IBM SPSS Statistics v25.0 software. Qualitative variables were summarized as frequencies (*n*) and percentages (%). Detection rates were compared using the chi-squared test for categorical variables. A *p*-value of <0.05 was considered statistically significant.

### 2.6. SLN Evaluation Parameters

The following SLN detection metrics were evaluated in this study:Detection rate (DR): the proportion of patients in whom at least one SLN was successfully identified.Bilateral detection rate (BDR): the proportion of patients with SLNs successfully identified on both sides of the pelvis.Sensitivity: the ratio of true-positive SLN identifications to the total number of patients with confirmed metastases.

## 3. Results

### 3.1. Patient Characteristics

The study cohort consisted of 119 patients with early-stage endometrial cancer. The median age was 60.8 years (range: 38–85). Detailed clinical and pathological characteristics are summarized in Table 1. The majority of patients had endometrioid histology (95%), while less common subtypes included serous carcinoma (2.5%) and clear-cell carcinoma (2.5%). Lymphovascular space invasion (LVSI) was present in 12 patients (10%). Myometrial invasion greater than 50% was observed in 36% of cases.

### 3.2. Sentinel Lymph Node Detection

The overall SLN DR was 97.5%. Bilateral detection was achieved in 86.5% of cases. DRs for individual and combined methods are detailed in Table 2 and illustrated in Figure 1.

### 3.3. Metastases and Sample Quality

Metastases to SLNs were identified in 14 patients (12%), including eight cases of unilateral metastases (57%), five cases of bilateral metastases (36%), and one case of isolated para-aortic node metastasis (7%) (Table 3, Figure 2).

Metastases to SLNs were identified using a single identification method in six patients: five with Tc99m and one with Patent Blue. An additional eight cases involved combined methods. On average, two SLNs were excised per patient. Notably, SLN samples identified using Tc99m showed no “empty nodes” (0%), compared to 1.7% with Patent Blue and 0.8% with ICG.

The overall detection rate of SLNs in this study was 97.5%. When analyzed by method, detection rates were as follows: 100% for ICG, 100% for Patent Blue, 96% for Tc99m, 97.3% for the combination of Patent Blue and Tc99m, and 96.9% for the combination of ICG and Tc99m. Bilateral SLN detection was achieved in 86.5% of cases, with individual bilateral detection rates of 85.7% for ICG, 91% for Patent Blue, 80% for Tc99m, 90.6% for the combination of ICG and Tc99m, and 86.5% for the combination of Patent Blue and Tc99m. These results align with the existing literature on both bilateral and unilateral SLN detection.

Histopathological analysis identified metastases in SLNs from 14 patients (12% of the cohort). Of these, eight patients (57%) had unilateral metastases, five (36%) had bilateral metastases, and one patient (7%) presented with an isolated para-aortic node metastasis. The metastases were classified as nine macrometastases, two micrometastases, and three cases of ITCs.

Metastases detected using a single identification method were found in six cases: five with Tc99m and one with Patent Blue. By contrast, combined methods detected metastases in eight patients: six with Tc99m and Patent Blue, and two with Tc99m and ICG. Notably, in two cases where Patent Blue was combined with Tc99m, SLNs were identified exclusively by Tc99m, as Patent Blue failed to stain the nodes.

Across all methods, the average number of SLNs excised per patient was consistent at two. The quality assessment of the excised SLN samples revealed differences in the presence of lymphatic tissue depending on the identification method. No “empty nodes” were observed with Tc99m (0%), while Patent Blue and ICG had rates of 1.7% (two cases) and 0.8% (one case), respectively.

## 4. Discussion

This study compared the effectiveness of different SLN identification methods in early-stage endometrial cancer. The findings demonstrate that combining technetium-99m (Tc99m) with dyes such as ICG or Patent Blue does not significantly improve SLN detection rates compared to using dyes alone. However, the combination enhances metastasis detection and reduces the incidence of “empty nodes”, thus improving the overall accuracy of the SLNB procedure.

### 4.1. SLN Identification Techniques

The cervical injection technique is widely recognized as the most reproducible and accessible method for SLN mapping in EC, given the minimal anatomical distortion of the cervix in most patients [16,17]. For instance, the FIRES trial demonstrated that standardized cervical injection of ICG resulted in successful SLN mapping in 86% of patients, with a positive node rate of 12% [18]. The effectiveness of SLN mapping depends not only on accurate detection but also on ensuring the excised nodes are the true first nodes in the lymphatic pathway and contain lymphatic tissue. Achieving bilateral mapping, with at least one SLN identified on each side, is a key objective, as it significantly improves metastasis detection compared to routine lymphadenectomy [19,20].

### 4.2. Detection Failures and the Issue of “Empty Nodes”

Failures in SLN detection remain a challenge, particularly with the use of ICG. Taskin et al. identified several factors potentially contributing to failed bilateral mapping, including longer uterine and cervical dimensions, deep myometrial invasion, and larger tumor size, although these factors were not statistically significant. In contrast, body mass index and injection volume were not associated with mapping failures [11]. Similarly, Sozzi et al. reported that bilateral SLN detection using ICG failed in 23.7% of cases, with lymphovascular space invasion (LVSI), non-endometrioid histology, and enlarged lymph nodes identified as significant risk factors [9]. However, in our study, LVSI did not appear to impact SLN detection rates, suggesting that differences in technique or population characteristics may influence outcomes.

ICG, despite its advantages, is associated with the issue of “empty nodes”, where nodes devoid of lymphatic tissue are mistakenly identified as SLNs. This phenomenon results from the albumin-binding properties of ICG, which increase oncotic pressure in lymphatic vessels, causing them to swell and be misinterpreted as lymph nodes [21]. Studies estimate that up to 40% of SLN detection failures with ICG are due to the removal of empty nodes. Early experiences reported failure rates as high as 20% during the first 25 procedures, which decreased to 7% as surgical expertise improved [21]. Additionally, ICG rapidly diffuses through the lymphatic system into second-tier nodes, complicating true SLN localization. A second injection improves bilateral detection rates to 96% [22].

The importance of surgical expertise and the learning curve in SLN mapping cannot be overstated. Both Khoury-Collado et al. and our findings highlight that detection rates improve significantly with experience, increasing from 77% to 94% after surgeons performed at least 30 procedures [23]. This underscores the critical role of training and experience in achieving reliable SLN identification and minimizing issues such as empty nodes.

Furthermore, as Holloway et al. emphasized, the integration of SLN mapping in EC not only provides a less invasive alternative to routine lymphadenectomy but also enhances the detection of lymph node metastases, including para-aortic nodes [20]. The adoption of advanced histopathological techniques, such as ultrastaging, further improves the detection rates of micrometastases and isolated tumor cells. These advancements highlight the pivotal role of SLN mapping in refining the accuracy of staging and guiding treatment decisions in endometrial cancer.

### 4.3. Strengths and Limitations of Individual Techniques

Patent Blue is a simple and cost-effective technique that does not require specialized equipment. The dye binds to serum proteins and migrates to SLNs within 5–10 min, staining them blue [20]. However, the SLN must be identified early in the procedure before the dye migrates further along the lymphatic vessels [23]. Additionally, severe allergic reactions, including anaphylaxis, have been reported in up to 1.9% of cases [24].

ICG provides deep tissue penetration and low autofluorescence, making it particularly effective in obese patients [20]. In the FIRES trial, ICG demonstrated mapping success rates comparable to those of Tc99m combined with blue dye [18]. Holloway et al. supported these findings, noting that ICG with near-infrared fluorescence imaging achieves mapping rates similar to radiocolloids while maintaining low false-negative rates [20].

Tc99m, as a radioisotopic tracer, offers deep tissue penetration and longer retention times in SLNs [25]. In our study, the use of Tc99m eliminated the issue of empty nodes, which occurred in 1.7% of cases with Patent Blue and 0.8% of cases with ICG. Preoperative SPECT/CT imaging with Tc99m provides superior spatial resolution compared to planar imaging, enhancing the accuracy of SLN localization [26,27]). Papadia et al. reported detection rates of 96.9% for the combination of Tc99m and ICG and 97.3% for Tc99m and Patent Blue, with ICG showing superiority in bilateral detection (84.1% vs. 73.5%, *p* = 0.007) [28].

The effectiveness of SLN detection is influenced by multiple technical factors and the operator’s expertise. A critical aspect is the administration technique, particularly the combination of superficial and deep injections into the cervix. This approach ensures better tracer distribution within the lymphatic system, enhancing detection rates for both dyes and radiotracers [21,23]. For dye-based methods, maintaining the appropriate time interval between injection and SLN removal is essential, as extended delays can result in dye diffusion to second-tier nodes, complicating localization [18,22].

The radioactive tracer Tc99m offers significant advantages in challenging conditions, such as during extensive manipulation of the retroperitoneal space. Under these circumstances, dyes often leak from lymphatic vessels, making SLN identification more difficult. Tc99m, with its stability and ability to be precisely localized using a gamma probe, provides a reliable solution even in such adverse scenarios [25,26]. Additionally, the presence of Tc99m in the excised node can be confirmed both intraoperatively and ex vivo, minimizing the risk of false identification [25,28].

### 4.4. Histopathological Evaluation and Ultrastaging

The adoption of ultrastaging has advanced the SLNB procedure by enabling the detection of low-volume metastases, including micrometastases and isolated tumor cells (ITCs). Serial sectioning and immunohistochemistry allow for more comprehensive node evaluation, with significant prognostic implications for treatment decisions [13]. Our findings underscore the value of advanced histopathological techniques in improving staging accuracy.

### 4.5. Study Limitations

This study has several limitations. The retrospective design and relatively small sample size may limit the statistical power of the findings. Additionally, while the study used standardized Tc99m protocols, variations in injection timing (“short” vs. “long” protocols) were not directly compared. The variability in surgical techniques and operator experience may also have influenced detection rates. Future prospective studies should include larger cohorts and formal power analyses to ensure robust statistical conclusions.

## 5. Conclusions

This study demonstrates that combining Tc99m with dyes such as ICG or Patent Blue does not significantly improve SLN detection rates compared to the use of dyes alone. However, this multimodal approach enhances the detection of metastases and minimizes the incidence of empty nodes, emphasizing its value in improving the accuracy of SLNB in endometrial cancer.

Our findings underscore the critical importance of standardized tracer protocols and advanced histopathological techniques, such as ultrastaging, in optimizing SLNB outcomes. These advancements not only improve diagnostic precision but may also influence subsequent treatment decisions. Although the retrospective nature of this study imposes certain limitations, it provides a valuable basis for designing future randomized prospective trials aimed at refining SLN detection protocols and establishing their broader clinical utility in the staging and management of endometrial cancer.

## Figures and Tables

**Figure 1 jcm-14-00676-f001:**
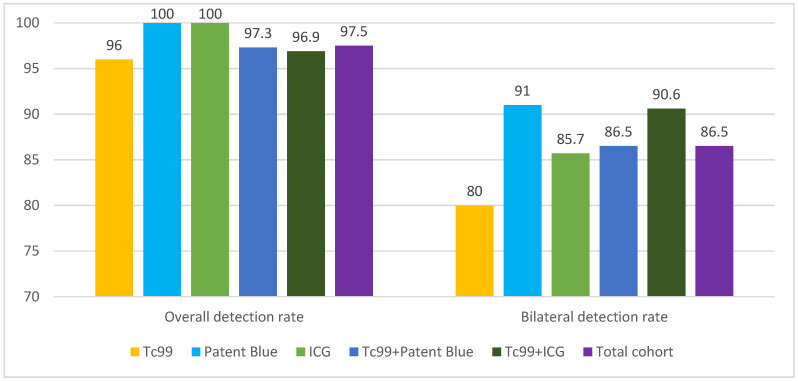
SLN Detection Rates by Method.

**Figure 2 jcm-14-00676-f002:**
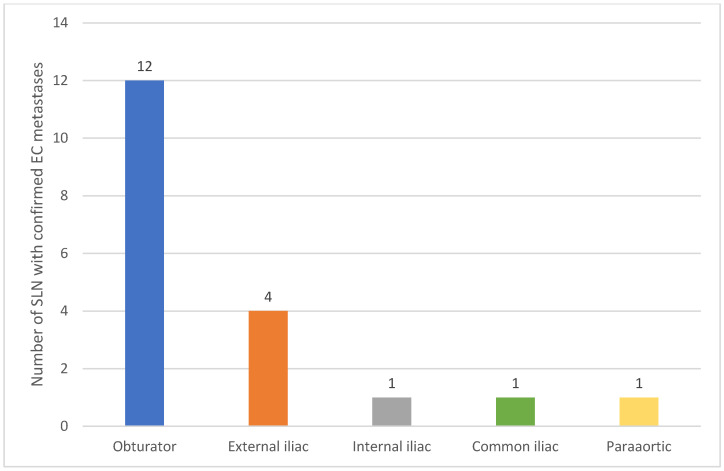
Distribution of SLN Metastases.

**Table 1 jcm-14-00676-t001:** Patient and Tumor Characteristics ^1^.

Feature	Characteristic	N	%
Histologic Type	Endometrioid	113	95.0%
	Serous	3	2.5%
	Clear-Cell	3	2.5%
Lymphovascular Space Invasion (LVSI)	Present	12	10.0%
	Absent	107	90.0%
Myometrial Invasion	0%	10	8.4%
	<50%	66	55.5%
	>50%	43	36.1%
Lymphadenectomy	Bilateral Pelvic	20	16.8%
	Paraaortic	7	5.9%
FIGO Stage	IA	11	9.2%
	IB	56	47.1%
	II	31	26.1%
	IIIA	3	2.5%
	IIIB	2	1.7%
	IIIC1	12	10.1%
	IIIC2	4	3.3%
FIGO Grade	G1	59	49.6%
	G2	52	43.7%
	G3	8	6.7%
Total		119	100.0%

^1^ Percentages may not sum to 100% due to rounding.

**Table 2 jcm-14-00676-t002:** Effectiveness of Sentinel Lymph Node (SLN) Detection ^1^.

Parameter	Tc99m (N = 25)	Blue Dye (N = 11)	ICG ^2^ (N = 14)	Tc99m + Blue Dye (N = 37)	Tc99m + ICG ^2^ (N = 32)	Total Cohort (N = 119)	*p*-Value
Overall Detection Rate	24 (96.0%)	11 (100.0%)	14 (100.0%)	36 (97.3%)	31 (96.9%)	116 (97.5%)	0.921
Bilateral Detection Rate	20 (80.0%)	10 (91.0%)	12 (85.7%)	32 (86.5%)	29 (90.6%)	103 (86.5%)	0.815
Confirmed Metastases	5 (25.0%)	1 (9.1%)	0 (0.0%)	6 (16.2%)	2 (6.3%)	14 (11.8%)	0.266

^1^ *p*-value < 0.05 was considered statistically significant. ^2^ ICG = indocyanine green.

**Table 3 jcm-14-00676-t003:** Location of Metastases in Sentinel Lymph Nodes ^1^.

Lymph Node Location	Unilateral Metastases (N = 8)	Bilateral Metastases (N = 5)	Isolated Metastasis (N = 1)
Obturator	5	7	0
External Iliac	2	2	0
Internal Iliac	1	0	0
Common Iliac	0	1	0
Para-aortic	0	0	1

^1^ Some patients had metastases in more than one location. N represents the number of patients.

## Data Availability

The original contributions presented in this study are included in the article. Further inquiries can be directed to the corresponding author(s).

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
