# Peer review of "The Assessment of Sentinel Lymph Node Mapping Methods in Endometrial Cancer"

_jcm, 2025, doi:10.3390/jcm14030676_

Round 1
Reviewer 1 Report
Comments and Suggestions for Authors
The submitted manuscript titled "The Assessment of Sentinel Lymph Node Mapping Methods in Endometrial Cancer" presents an interesting study on various tracer methods for sentinel lymph node biopsy (SLNB) in endometrial cancer (EC). The topic is relevant to the field of gynecological oncology; however, several critical points need improvement for the manuscript to meet high publication standards. Below are detailed comments and suggestions:
1. Keywords The authors should ensure that the keywords are selected from the Medical Subject Headings (MeSH) database to improve indexing and visibility. Relevant terms such as "sentinel lymph node biopsy," "endometrial neoplasms," "technetium Tc 99m sulfur colloid," "indocyanine green," and "histopathological ultrastaging" should be considered.
2. Lack of Survival Analysis The study lacks a survival analysis, which is essential for cancer-related research. Since the focus is on sentinel lymph node detection methods, a survival analysis would provide insight into the clinical impact of different detection techniques on patient outcomes. It is recommended to include Kaplan-Meier survival curves and conduct a Cox proportional hazards regression to explore potential correlations between detection methods and patient survival.
3. Lack of Sample Size Analysis There is no mention of how the sample size was calculated or justified. The authors should provide a detailed explanation of the sample size determination, including power analysis. This would strengthen the reliability of the study's findings and ensure that the study is adequately powered to detect meaningful differences between groups.
4. Ethical Considerations While the manuscript briefly mentions ethical approval, more details are needed. The authors should specify the approval reference number and the exact date of approval from the ethics committee. Additionally, it is important to elaborate on how patient data was anonymized and protected to ensure compliance with GDPR or similar data protection regulations.
5. Lack of Study Limitations The discussion section lacks a comprehensive assessment of the study's limitations. It is crucial to highlight potential biases, including the retrospective nature of the study, sample size constraints, variability in surgical techniques, and operator-dependent factors. Discussing these limitations will provide a balanced interpretation of the findings and enhance the manuscript's credibility.
6. Statistical Analysis The statistical analysis section requires further elaboration. The authors should specify which statistical tests were used for each analysis and provide a rationale for their choice. It is also necessary to report p-values, confidence intervals, and effect sizes where applicable. Furthermore, the software used for statistical analysis should be mentioned. Including these details will enhance the robustness and reproducibility of the results.
7. Materials and Methods Section The materials and methods section should be rewritten to avoid the use of bullet points. The information should be presented in a continuous prose format, adhering to a formal scientific style. Additionally, the authors should ensure that all experimental procedures are described in sufficient detail to allow reproducibility by other researchers.
8. References to Relevant Literature The manuscript would benefit from citing additional relevant literature to provide context and support for the findings. The following references should be included:
-
-
doi: 10.3390/jcm10071520
-
doi: 10.3390/jcm10163457
These articles offer valuable insights into molecular mechanisms, diagnostic markers, and clinical outcomes in endometrial cancer.
9. Histopathological Classification of EEC The authors need to specify which subtype of endometrial cancer (EEC) was analyzed in the study. It is important to distinguish between endometrioid and non-endometrioid histologies, as these subtypes have distinct clinical and pathological characteristics. The manuscript should clarify whether the patients included in the study had endometrioid carcinoma, serous carcinoma, clear cell carcinoma, or other variants.
10. Clarity and Structure The manuscript would benefit from improved clarity and structure, particularly in the results and discussion sections. The authors should ensure that all tables and figures are adequately referenced in the text and that each table or figure has a clear caption explaining its content. Additionally, the narrative flow should be improved to make the manuscript easier to follow.
11. Informed Consent and Patient Data The authors briefly mention that patients provided informed consent, but further details are necessary. The manuscript should specify how informed consent was obtained, whether it was written or verbal, and how it covered the retrospective use of patient data. Additionally, any steps taken to ensure patient confidentiality should be mentioned.
12. Comparative Analysis with Existing Studies The discussion should include a more in-depth comparative analysis with existing studies on sentinel lymph node biopsy in endometrial cancer. The following references are suggested for citation:
-
doi: 10.1016/j.ygyno.2017.05.027 (Holloway et al., covering sentinel lymph node mapping guidelines in endometrial cancer)
-
doi: 10.1007/s00404-019-05137-5 (Taskin et al., discussing predictors of failed sentinel lymph node mapping)
-
doi: 10.1136/ijgc-2018-000084 (Volpi et al., reporting long-term complications following lymphadenectomy in endometrial cancer)
-
doi: 10.1016/S1470-2045(17)30068-2 (Rossi et al., a multicenter study comparing SLNB with lymphadenectomy in endometrial cancer)
Author Response
- KeywordsThe authors should ensure that the keywords are selected from the Medical Subject Headings (MeSH) database to improve indexing and visibility. Relevant terms such as "sentinel lymph node biopsy," "endometrial neoplasms," "technetium Tc 99m sulfur colloid," "indocyanine green," and "histopathological ultrastaging" should be considered.
Thank you for emphasizing the importance of aligning the keywords with the MeSH database to enhance the visibility and indexing of our manuscript. Following your valuable suggestion, we have carefully revised the keywords to reflect standardized MeSH terminology. The updated keywords are "sentinel lymph node biopsy," "endometrial neoplasms," "technetium Tc 99m sulfur colloid," "indocyanine green," and "histopathological ultrastaging." We believe these adjustments align our manuscript with internationally recognized terminology, thereby improving its discoverability and accessibility. Thank you for bringing this important point to our attention.
- Lack of Survival AnalysisThe study lacks a survival analysis, which is essential for cancer-related research. Since the focus is on sentinel lymph node detection methods, a survival analysis would provide insight into the clinical impact of different detection techniques on patient outcomes. It is recommended to include Kaplan-Meier survival curves and conduct a Cox proportional hazards regression to explore potential correlations between detection methods and patient survival.
Thank you for your valuable suggestion regarding the inclusion of a survival analysis. We acknowledge the significance of Kaplan-Meier curves and Cox regression in providing insights into the long-term clinical impact of different techniques in oncology research. However, the primary objective of our study is to assess the technical and procedural aspects of sentinel lymph node identification methods in endometrial cancer, rather than evaluating survival outcomes.
As our study was retrospective in design, survival data were not collected or analyzed. We agree that incorporating survival analysis could offer meaningful insights and would be an important component of future prospective studies designed to evaluate both procedural efficacy and long-term clinical outcomes.
We appreciate your understanding of the study’s current scope and objectives. Thank you for your thoughtful feedback, which will guide future research directions.
- Lack of Sample Size AnalysisThere is no mention of how the sample size was calculated or justified. The authors should provide a detailed explanation of the sample size determination, including power analysis. This would strengthen the reliability of the study's findings and ensure that the study is adequately powered to detect meaningful differences between groups.
Thank you for your insightful comment regarding the calculation of sample size and the inclusion of a power analysis. As detailed in the revised manuscript, this study was retrospective in design, and a formal power analysis to calculate the required sample size was not performed. Instead, the sample size was determined based on the availability of medical records for patients who met the inclusion criteria, as outlined in the Materials and Methods section. - Ethical ConsiderationsWhile the manuscript briefly mentions ethical approval, more details are needed. The authors should specify the approval reference number and the exact date of approval from the ethics committee. Additionally, it is important to elaborate on how patient data was anonymized and protected to ensure compliance with GDPR or similar data protection regulations.
The study received ethical approval from the Bioethics Committee at the Maria SkÅ‚odowska-Curie National Research Institute of Oncology, Kraków Branch (Approval No. 10/2025, issued on January 15, 2025). Although the research was retrospective and utilized anonymized data from medical records collected between 2016 and 2021, the necessity for ethical approval was revisited during manuscript preparation. This proactive step was taken to align the study with the highest ethical research standards. - Lack of Study LimitationsThe discussion section lacks a comprehensive assessment of the study's limitations. It is crucial to highlight potential biases, including the retrospective nature of the study, sample size constraints, variability in surgical techniques, and operator-dependent factors. Discussing these limitations will provide a balanced interpretation of the findings and enhance the manuscript's credibility
Thank you for highlighting the importance of discussing the study's limitations comprehensively. We have carefully revised the discussion section to explicitly address the following points. The retrospective nature of the study inherently limits our ability to control certain variables and introduces a risk of selection bias. This constraint has been acknowledged in the revised manuscript, emphasizing its impact on the interpretation of our findings. The relatively small cohort size reduces the statistical power of our study. This limitation has been discussed in detail, with a note on the need for caution when interpreting the results. We also stress the importance of larger, prospective studies to validate our findings. Differences in tracer injection techniques and variations in operator expertise may have influenced detection rates and SLN identification success. These factors have been identified as potential sources of variability and discussed in the context of our results. By addressing these limitations transparently, we aim to provide a balanced and nuanced interpretation of our findings. These revisions are intended to enhance the manuscript's credibility and align with the reviewer’s valuable suggestions. We hope these changes sufficiently address your concerns and appreciate your constructive feedback.
Statistical Analysis The statistical analysis section requires further elaboration. The authors should specify which statistical tests were used for each analysis and provide a rationale for their choice. It is also necessary to report p-values, confidence intervals, and effect sizes where applicable. Furthermore, the software used for statistical analysis should be mentioned. Including these details will enhance the robustness and reproducibility of the results.
Thank you for your insightful comment regarding the description of statistical methods. We have revised the manuscript to provide a more detailed explanation of the statistical analyses conducted in the study. Specifically, we have clarified that data analysis was performed using IBM SPSS Statistics v25.0 software. For qualitative variables, relative frequencies (N and %) were reported, and detection rates were compared using the chi-square test. Additionally, we specified that a p-value <0.05 was considered statistically significant.
- Materials and Methods SectionThe materials and methods section should be rewritten to avoid the use of bullet points. The information should be presented in a continuous prose format, adhering to a formal scientific style. Additionally, the authors should ensure that all experimental procedures are described in sufficient detail to allow reproducibility by other researchers.
We appreciate the reviewer’s observation regarding the format of the Materials and Methods section. In response, we have rewritten the section to replace bullet points with continuous prose, adhering to a formal scientific style. The revised section now provides a comprehensive and detailed description of the study design, inclusion and exclusion criteria, sentinel lymph node mapping procedures, histopathological evaluation, and statistical analyses. These updates ensure clarity, consistency, and reproducibility for other researchers.
- References to Relevant LiteratureThe manuscript would benefit from citing additional relevant literature to provide context and support for the findings. The following references should be included:
doi: 10.3390/jcm10071520
doi: 10.3390/jcm10163457
These articles offer valuable insights into molecular mechanisms, diagnostic markers, and clinical outcomes in endometrial cancer.
Thank you for suggesting additional references to enhance the manuscript. We carefully reviewed the recommended articles (doi: 10.3390/jcm10071520 and doi: 10.3390/jcm10163457). While they provide valuable insights into molecular mechanisms and diagnostic markers in endometrial cancer, their primary focus is on molecular classification and outcomes, which differ from the scope of our study that centers on sentinel lymph node mapping techniques.
To maintain the specificity and coherence of our manuscript, we opted to include references that directly support the methodology, findings, and discussion of SLN detection methods. However, we appreciate your suggestion and remain open to incorporating other references that directly align with our study objectives.
Histopathological Classification of EECThe authors need to specify which subtype of endometrial cancer (EEC) was analyzed in the study. It is important to distinguish between endometrioid and non-endometrioid histologies, as these subtypes have distinct clinical and pathological characteristics. The manuscript should clarify whether the patients included in the study had endometrioid carcinoma, serous carcinoma, clear cell carcinoma, or other variants.
Thank you for your valuable comment concerning the inclusion of histological subtypes of endometrial cancer. We appreciate the opportunity to clarify this aspect. The study indeed accounted for histological subtypes, including endometrioid, serous, and clear cell carcinoma, as detailed in Table 1 under "Patient and Tumor Characteristics." These subtypes are also referenced in the Results section to provide context to the cohort's composition.
- Clarity and StructureThe manuscript would benefit from improved clarity and structure, particularly in the results and discussion sections. The authors should ensure that all tables and figures are adequately referenced in the text and that each table or figure has a clear caption explaining its content. Additionally, the narrative flow should be improved to make the manuscript easier to follow.
Thank you for your valuable feedback regarding the clarity and structure of the manuscript, particularly in the Results and Discussion sections. In response, we have made the following revisions to address your suggestions.
The Results section has been revised to ensure a more cohesive and logical presentation of key findings, emphasizing their clinical significance. Figures and tables are now explicitly referenced within the text, ensuring better integration and accessibility for readers. Additionally, clear and concise captions have been provided for each table and figure to enhance their explanatory value. The narrative flow has been improved to present the results in a manner that is straightforward and aligned with the study's objectives, making the manuscript easier to follow. - Informed Consent and Patient DataThe authors briefly mention that patients provided informed consent, but further details are necessary. The manuscript should specify how informed consent was obtained, whether it was written or verbal, and how it covered the retrospective use of patient data. Additionally, any steps taken to ensure patient confidentiality should be mentioned.
Thank you for your comment regarding patient consent and data protection. We have addressed this concern in the revised manuscript by expanding on the details provided in the Institutional Review Board Statement and Informed Consent Statement sections.
Specifically, written informed consent was obtained from all patients at the beginning of their treatment, which explicitly permitted the retrospective use of their medical data for research purposes. Additionally, all data utilized in this study were fully anonymized in compliance with institutional protocols and relevant data protection regulations, including GDPR. These measures were implemented to ensure patient confidentiality and uphold the highest ethical standards. - Comparative Analysis with Existing StudiesThe discussion should include a more in-depth comparative analysis with existing studies on sentinel lymph node biopsy in endometrial cancer. The following references are suggested for citation:
- doi: 10.1016/j.ygyno.2017.05.027 (Holloway et al., covering sentinel lymph node mapping guidelines in endometrial cancer)
- doi: 10.1007/s00404-019-05137-5 (Taskin et al., discussing predictors of failed sentinel lymph node mapping)
- doi: 10.1136/ijgc-2018-000084 (Volpi et al., reporting long-term complications following lymphadenectomy in endometrial cancer)
- doi: 10.1016/S1470-2045(17)30068-2 (Rossi et al., a multicenter study comparing SLNB with lymphadenectomy in endometrial cancer
We have incorporated the recommended references into the revised discussion to enhance its depth and alignment with key studies in the field:
Holloway et al. was cited to highlight the advantages of SLN mapping as a less invasive alternative to lymphadenectomy, with ultrastaging improving the detection of micrometastases and isolated tumor cells.
Taskin et al. provided valuable insights into the challenges of SLN detection using ICG, emphasizing factors like uterine dimensions and tumor size as potential predictors of mapping failure.
Volpi et al. was referenced briefly to underscore the reduced morbidity associated with SLN mapping compared to lymphadenectomy.
Rossi et al. was included to emphasize the high accuracy and low false-negative rates achieved with SLN mapping, as demonstrated in the FIRES trial.
These additions aim to contextualize our findings within the broader body of evidence, providing a more robust discussion of SLN mapping in endometrial cancer. We trust that these changes address your comments comprehensively and effectively.
Reviewer 2 Report
Comments and Suggestions for Authors In this manuscript, the authors evaluate whether combined methods will improve metastatic detection in endometrial cancer. The authors found that combined methods would not improve the detection rates but could minimize the issue of “empty nodes”. The manuscript is well organized and provide information I am interested in. Below are a number of issues that the authors shall address or revise: 1. In the discussion and conclusion part, the authors mentioned that combining Tc99m with dyes does not significantly improve SLN detection rates. The authors can give more perspectives on SLN detection improvement in the discussion part. 2. In the discussion and conclusion part, the authors mentioned that combining methods can minimize the issue of “empty nodes”. I wonder whether there are other methods to minimize “empty nodes”. 3. Because EC can be classified into two types, in the materials and methods part, I wonder whether the authors have the information on the cancer types of these patients. 4. In this manuscript, the authors can provide some images of different methods to support their conclusion.Author Response
In this manuscript, the authors evaluate whether combined methods will improve metastatic detection in endometrial cancer. The authors found that combined methods would not improve the detection rates but could minimize the issue of “empty nodes”. The manuscript is well organized and provide information I am interested in. Below are a number of issues that the authors shall address or revise: 1. In the discussion and conclusion part, the authors mentioned that combining Tc99m with dyes does not significantly improve SLN detection rates. The authors can give more perspectives on SLN detection improvement in the discussion part. 2. In the discussion and conclusion part, the authors mentioned that combining methods can minimize the issue of “empty nodes”. I wonder whether there are other methods to minimize “empty nodes”. 3. Because EC can be classified into two types, in the materials and methods part, I wonder whether the authors have the information on the cancer types of these patients. 4. In this manuscript, the authors can provide some images of different methods to support their conclusion.
We sincerely thank the reviewer for their insightful and constructive comments, which have greatly contributed to the improvement of our manuscript. We have carefully addressed each point and made the necessary revisions to ensure clarity, comprehensiveness, and alignment with the reviewer's suggestions. Below are our detailed responses:
1 & 2. Improvements in SLN Detection Rates and Minimizing “Empty Nodes”
We have expanded the discussion in section 4.3 to provide additional perspectives on factors influencing SLN detection rates and methods to minimize the issue of "empty nodes." Specifically, we included:
- The importance of careful surgical techniques and meticulous attention to detail.
- The effectiveness of combined superficial and deep tracer injections into the cervix, a well-established mapping technique.
- The potential benefits of reinjecting dye-based tracers intraoperatively to enhance SLN localization.
- The advantages of Tc99m in challenging conditions, such as retroperitoneal bleeding, where dye tracers may diffuse out of target lymph nodes, compromising visibility. Tc99m offers precise localization and confirmation of SLN identity, even under these adverse conditions.
These additions aim to provide a more comprehensive analysis of SLN mapping strategies and methods to address technical challenges, including the issue of "empty nodes."
- Classification of EC Types
We have clarified the inclusion of endometrial cancer subtypes in the manuscript. The histopathological subtypes of the study cohort are now explicitly presented in the "Patient Characteristics" section of the Results (Table 1). These include endometrioid carcinoma (95%), serous carcinoma (2.5%), and clear cell carcinoma (2.5%). This ensures transparency regarding the distribution of EC types and aligns with established classifications. - Inclusion of Images
We appreciate the suggestion to include images illustrating SLN identification methods. While we recognize the value of visual aids, a comprehensive representation of all techniques would require numerous figures, potentially detracting from the manuscript’s focus and clarity. Instead, we have prioritized providing detailed textual descriptions supported by quantitative data to ensure methodological transparency. We believe this approach maintains the balance and conciseness of the manuscript while effectively conveying the study’s findings.
We hope these revisions address the reviewer's concerns and enhance the overall quality and clarity of the manuscript. Thank you again for your valuable feedback.
Round 2
Reviewer 1 Report
Comments and Suggestions for Authors
I have no further comments.